# Development and validation of an AI algorithm to generate realistic and meaningful counterfactuals for retinal imaging based on diffusion models

**Indu Ilanchezian**[1,2]☯, **Valentyn Boreiko**[2,3]☯, **Laura Kühlewein**[4] (ID), **Ziwei Huang**[1,2], **Murat Seçkin Ayhan**[5], **Matthias Hein**[2,3], **Lisa Koch**[1,2,6]*, **Philipp Berens** (ID)[1,2]*

**1** Hertie Institute for AI in Brain Health, University of Tübingen, Tübingen, Germany, **2** Tübingen AI Center, Tübingen, Germany, **3** Department of Computer Science, University of Tübingen, Tübingen, Germany, **4** Center for Ophthalmology, Institute for Ophthalmic Research, University of Tübingen, Tübingen, Germany, **5** Institute of Ophthalmology, University College London, London, United Kingdom, **6** Department of Diabetes, Endocrinology, Nutritional Medicine and Metabolism UDEM, Inselspital, Bern University Hospital, University of Bern, Bern, Switzerland

☯ These authors contributed equally to this work.
* lisa.koch@uni-tuebingen.de (LK); philipp.berens@uni-tuebingen.de (PB)

**Data availability statement:** All code needed to replicate the study's findings are available on

## Abstract

Counterfactual reasoning is often used by humans in clinical settings. For imaging based specialties such as ophthalmology, it would be beneficial to have an AI model that can create counterfactual images, illustrating answers to questions like "If the subject had had diabetic retinopathy, how would the fundus image have looked?". Such an AI model could aid in training of clinicians or in patient education through visuals that answer counterfactual queries. We used large-scale retinal image datasets containing color fundus photography (CFP) and optical coherence tomography (OCT) images to train ordinary and adversarially robust classifiers that classify healthy and disease categories. In addition, we trained an unconditional diffusion model to generate diverse retinal images including ones with disease lesions. During sampling, we then combined the diffusion model with classifier guidance to achieve realistic and meaningful counterfactual images maintaining the subject's retinal image structure. We found that our method generated counterfactuals by introducing or removing the necessary disease-related features. We conducted an expert study to validate that generated counterfactuals are realistic and clinically meaningful. Generated color fundus images were indistinguishable from real images and were shown to contain clinically meaningful lesions. Generated OCT images appeared realistic, but could be identified by experts with higher than chance probability. This shows that combining diffusion models with classifier guidance can achieve realistic and meaningful counterfactuals even for high-resolution medical images such as CFP images. Such images could be used for patient education or training of medical professionals.

Github [https://github.com/berenslab/retinal_
image_counterfactuals]. Color fundus images
were obtained from EyePacs Inc., a third-party
provider, through a Diabetic Retinopathy (DR)
screening program and cannot be publicly
shared. Data are available from EyePacs Inc.
(contact@eyepacs.org) upon request for a fee.
For OCT, we used the publicly available data set
from https://data.mendeley.com/datasets/
rscbjbr9sj/3.

**Funding:** This work was supported the German
Ministry of Science and Education (01IS18039A
to PB); the German Research Foundation
(BE5601/8-1 to PB; EXC 2064 - Project number
390727645 to PB; INST 37/1057-1 to MH), the
Carl Zeiss Foundation (project "Certification and
Foundations of Safe Machine Learning Systems
in Healthcare" to MH and LK) and the Hertie
Foundation (to PB). PB is a member of the Else
Kröner Medical Scientist Kolleg "ClinbrAIn:
Artificial Intelligence for Clinical Brain
Research". The funders had no role in study
design, data collection and analysis, decision to
publish, or preparation of the manuscript.

**Competing interests:** The authors have
declared that no competing interests exist.

## Introduction

Humans naturally use counterfactual thoughts, deliberations and statements to reason about the causal structure of the world, understand the past and prepare for the future [1]. For example, counterfactuals are used in medicine to explain decisions or weigh alternatives: "If we had treated the patient with drug X, she might have experienced severe side effects."[2]. In a similar way, when medical images are available, it may be useful to create counterfactual images that visualize the answer to the question [3]: "For a given subject who we believe is healthy, how would the imaging data have looked for the same subject to be identified as the diseased class?", e.g. for training of medical professionals or educating patients about consequences of actions.

In ophthalmology, for instance, clinicians regularly use imaging modalities such as retinal fundus photography and Optical Coherence Tomography (OCT) for diagnosing sight-threatening diseases like Diabetic Retinopathy (DR) and Age-Related Macular Degeneration (AMD). Counterfactual images as described above can be generated from Deep Neural Networks (DNNs) that are trained to detect the presence of these diseases [4]. In this context, counterfactuals are artificially generated images that contain minimal, realistic and meaningful changes to an input image such that the DNN classifier alters its decision to a desired target class [5]. For them to look realistic and meaningful, the model used to create them needs to have outstanding generative abilities. The resulting images can then also be viewed as explanations of the DNN's decisions as they enable the user of a DNN model to visualise the features that the classifier relies on for detecting the disease [6,7].

Previously, different strategies for generating counterfactuals have been proposed [5–12]. For example, DNN-based counterfactuals can be generated by iteratively superimposing the input image with the gradients of an adversarially robust classifier, which has more informative gradients than a plain model [5,6,13]. While these so-called sparse counterfactuals show meaningful features, they appear to modify the original image in unexpected and unnatural ways. On fundus images, they cover lesions with unnatural blood vessels in order to generate healthy counterfactuals [6]. In a similar vein, GAN-style models have been used for generation of lesions on fundus images [14]. For example, GANs have been used to synthesize cropped fundus images around the optic disc with signs of glaucoma, which focus on only part of the fundus image [15]. Furthermore, they have been used to generate clinically meaningful lesions on fundus images , requiring additional models to keep the vessel structure and optic disc intact [16] or generate the lesions [17]. These models are not suitable for our goal of generating counterfactual images, however, as the resulting images are often not realistic as they either contain unrealistic vessel structure [17] or fail to transfer the style and color of the original image to the generated images [16]. This is likely because the latent space of GANs intertwines patient attributes with other variables such as camera type, image quality or illumination level [18]. Finally, StyleGANs have been used to generate counterfactual retinal fundus images for a Diabetic Macular Edema classifier [8], but the counterfactuals generated with this procedure begin to show features relevant to target class even before the decision of the classifier changes, despite their highly realistic appearances.

Alternatively, denoising diffusion probabilistic models have been investigated for the generation of retinal fundus images [19], however, these methods focus on generation of healthy retinal fundus images rather than counterfactual image generation and were validated only for lower resolution images. Counterfactuals of OCT scans have also been generated using GANs to study retinal aging, but domain experts were easily able to identify the generated images [10], suggesting that they are not sufficiently realistic. Finally, in other medical domains such as brain tumor detection from MRI images and chest X-ray interpretation, counterfactuals

based on diffusion models have been used to generate healthy counterfactuals from diseased images [11,12], but not for generating images showing a disease from healthy ones.

Here, we show that we can generate realistic counterfactual fundus images for diabetic retinopathy by relying on deep generative models known as diffusion models [4]. These diffusion models have been shown to outperform GANs in realistic image generation, while also overcoming their drawbacks by producing diverse samples and covering a broad range of the image distribution in tandem with a stable training process [20,21]. We used classifiers trained to detect diabetic retinopathy from fundus images and then show how to combine these with a generative diffusion model to result in realistic counterfactual retinal images from healthy to diseased and vice versa. Importantly, we show (a) that domain specialists – ophthalmologists and AI researchers – view the resulting images as realistic when probed in an odd-one-out task and that (b) the induced changes are meaningful, as counterfactual images can be classified as well as real images. We also show that the framework generalizes to generating OCT B-scan counterfactuals for a diabetic macular edema, drusen and choroidal neovascularizations, although experts are still able to discriminated generated counterfactuals from real OCT B-scans. Overall, this indicates that our method is an important step towards generating counterfactual images that fulfill the criteria for counterfactual images to be used in medical reasoning as outlined above, towards realizing the clinical potential of counterfactual AI models [3].

## Methods

We first describe the retina imaging datasets used in this study and then review the relevant methods for the generation of counterfactuals for such images. Lastly, we describe our design of a user study in order to evaluate the clinical relevance of counterfactuals.

### Datasets

We used retina imaging data sets from two common modalities: (1) color fundus photography (CFP) and (2) Optical Coherence Tomography (OCT).

Color fundus images were obtained from EyePacs Inc., a third-party provider, through a Diabetic Retinopathy (DR) screening program. Data are available from EyePacs Inc. (contact@eyepacs.org) upon request for a fee. Initially, this collection contained over $180,000$ retinal fundus images from over $42,000$ subjects along with meta data such as age, sex, race and blood pressure. Image quality was indicated as "Insufficient for Full Interpretation", "Adequate", "Good" or "Excellent" per image as annotated by Eyepacs Inc. Some DR labels were missing. We used "Good" and "Excellent" quality images with DR labels only, resulting in $92,745$ retinal fundus images from $27,926$ participants. Then, we created training, validation and test splits subject-wise to ensure that all images from a single participant were assigned to the same set (see Table 1). The training set was augmented with 789 images from the Benitez data set [22] and 1842 images from the FGADR data set [23] in order to strengthen the representation of diseased samples for the diffusion models. No additional quality control was performed for these datasets. All images were cropped to square dimensions of $224 \times 224$ pixels using a circle fitting procedure (https://github.com/berenslab/fundus_circle_cropping/tree/v0.1.0).

For OCT, we used a publicly available data set consisting of a total of $108,309$ images belonging to one of four categories [24] : normal, choroidal neovascularization (CNV), drusen and Diabetic Macular Edema (DME) (Table 1). This data set consists of OCT B-scans of the foveal region [24]. As the scans recorded by different devices had varying image resolutions, we standardized the image sizes by performing a square crop. We used only the images

**Table 1. Summary of the retinal image collections used for model development and evaluation.** *Datasets* used for training classifiers (both plain and adversarially robust ones) are marked with superscript c, and for the diffusion models - with superscript d.

| | | | | Training | Validation | Test |
|---|---|---|---|---|---|---|
| CFP | EyePacs[c,d] | subjects | | 15,827 | 5,324 | 6,775 |
| | | images | all | 46,921 | 15,658 | 30,166 |
| | | | healthy | 38,502 | 12,748 | 24,627 |
| | | | mild | 3,244 | 1,163 | 2,378 |
| | | | moderate | 4,695 | 1,572 | 2,907 |
| | | | severe | 238 | 121 | 127 |
| | | | proliferative | 242 | 54 | 127 |
| | Benitez[d] | images | all | 789 | - | - |
| | | | healthy | 94 | - | - |
| | | | mild | 6 | - | - |
| | | | moderate | 102 | - | - |
| | | | severe+ | 587 | - | - |
| | FGADR[d] | images | all | 1,842 | - | - |
| | | | healthy | 101 | - | - |
| | | | mild | 212 | - | - |
| | | | moderate | 595 | - | - |
| | | | severe+ | 934 | - | - |
| OCT | Kermany[c,d] | subjects | | 3558 | 712 | 474 |
| | | images | all | 71,231 | 14,714 | 10,496 |
| | | | normal | 34,340 | 6,813 | 4,464 |
| | | | CNV | 23,133 | 5,091 | 3,738 |
| | | | drusen | 5,393 | 1,221 | 1,288 |
| | | | DME | 8,365 | 1,589 | 1,006 |

with size $496 \times 512$ and $496 \times 768$ (96, 441 scans) as with these resolutions we could retain a larger portion of the image after the square crop. We created training, validation and test splits again subject-wise with 75% subjects in training, 15% in validation and 10% in the test set, respectively (see Table 1).

## Generating realistic counterfactual retinal images

We define visual counterfactuals as minimal, realistic and high-confidence changes to an image $x_0$ by which a classifier's prediction can be altered to a desired target class [5]. These images illustrate what features are important for the classifier to change the decision to a particular class, reflecting presumably the disease concept. Since the generative capabilities of a classifier are typically limited and it cannot by itself generate realistic counterfactuals, we rely on a diffusion model [21] to achieve realism. In order to generate counterfactuals, the reverse diffusion process is modified such that classifier gradients contribute to this process and guide the diffusion model towards producing counterfactuals in the desired class [25].

## Diffusion models

Diffusion models [20,21] are generative image models that yield high quality realistic images from the data distribution on which they have been trained. Diffusion models are driven by bi-directional forward and reverse processes that run for $T$ time steps where $T$ is sufficiently large. In the forward process, noise with subsequently increasing strength based on a schedule is added to images from the data distribution such that at the final time step $T$ the resulting distribution approaches the standard normal distribution. In the reverse process, one starts with a noisy image from the standard normal distribution at time step $T$ and passes it through

each time step in reverse thereby denoising it and arriving at a sample image from the original data distribution at time step 0. Each step in the reverse direction requires a transition probability distribution from a noisy image to a less noisy image, which is intractable to compute in practice. Hence, the mean and variance of these distributions are approximated using a neural network that is shared across the different time steps. Once the neural network is trained using the original data distribution, one can generate endless number of samples from the diffusion model by randomly sampling from the standard normal distribution and passing it through the reverse process. Such a sampling procedure is termed as unconditional as the model generates any sample from the data distributions without any constraints (S1 Text).

For both fundus and OCT data sets, we trained the diffusion model $p_\theta$ for $300,000$ minibatch iterations unconditionally with $1,000$ time steps and a linear noise schedule for the diffusion process. The diagonal covariance $\Sigma_\theta$ are also learned by the model during training. We used the same diffusion model architecture and parameters that are recommended in Open AI's guided diffusion repository (https://github.com/openai/guided-diffusion, [21]) for the $256 \times 256$ unconditional case. For the fundus data set, we additionally balanced the data set by oversampling the diseased classes to have a equal representation as that of the healthy class.

## Plain and adversarially robust classifiers

The sampling procedure described above results in unconditional samples from the data distribution whereas counterfactuals must belong to a specified target class, thus requiring conditional sampling. Classifiers can be used to drive diffusion models towards producing realistic images that belong to a desired class [21,26,27]. More specifically, the gradients of a classifier with respect to the image shift the mean of the reverse transition probabilities to guide the diffusion model in the right direction. Nevertheless, a plain classifier does not have gradients that are perceptually aligned with the features of a particular class and could result in counterfactuals that look visually similar to the original image when the changes are constrained to be minimal. In contrast, the gradients of adversarially robust models have strong generative properties and are more effective in guiding diffusion models towards generating meaningful features for a target class [13,25] despite being subjected to a constraint for producing minimal changes.

This property of adversarially robust models can be attributed to their training procedures which expose them to adversarial attacks. Consider a $K$-class classifier $f_\phi$ with parameters $\phi$, logits $f_\phi(x) \in \mathbb{R}^K$ and output probabilities $p_\phi(c|x) \in [0,1]^K$ where $x \in \mathbb{R}^d$ is the input to the classifier and $c \in \{1, \dots, K\}$. A targeted adversarial attack adds imperceptible perturbations to a starting image $x_0$ which changes the decision of the classifier from the correct class to a target class $k$ (for details, see S1 Text).

To defend the classifier $f_\phi$ empirically against such attacks, one can perform adversarial training. A well-known and commonly used algorithm for this is TRADES [28] algorithm whose loss function incorporates a term for the adversarial examples in addition to the standard cross-entropy loss. This term penalizes the KL divergence between the class-probability distributions of the classifier at the original image and at a corresponding adversarially perturbed image (for details, see S1 Text).

For retinal fundus images, we trained both plain and adversarially robust classifiers in binary and multi-class settings. In the multi-class setting, the task is a 5-way classification among the classes "healthy", "mild", "moderate", "severe" and "proliferative". In the binary setting, disease onset is considered from the "moderate" class, hence, "healthy" and "mild" are grouped into the normal category and the other classes to the diseased category. For OCT scans, we trained both plain and adversarially robust classifiers in the multi-class setting to

classify among the classes "healthy", "choroidal neovascularization (CNV)", "drusen" and "diabetic macular edema (DME)".

All plain and robust classifiers were ResNet-50 models trained for 100 epochs with an SGD optimizer with a learning rate of 0.01 and a cosine learning rate schedule. The fundus plain classifiers were initialized with weights from ImageNet pre-trained models and the fundus robust classifier with weights from a robustly pre-trained ImageNet model [29], after evaluating different training settings (see S1 Table). All OCT classifiers were initialized with random weights. We used the cross-entropy (CE) loss as objective function for the plain model and the TRADES loss [28] with $\varepsilon = 0.01$ for the fundus robust classifiers and $\varepsilon = 0.5$ for the OCT classifier. For both cases, we used $p = 2$.

## Diffusion visual counterfactuals

Here, we describe how to produce realistic Diffusion Visual Counterfactuals (DVCs). Following [25], we combined an unconditionally trained diffusion model $p_\theta$ as described above with an independently trained classifier $f_\phi$ so that the diffusion model can generate class-conditional samples.

Intuitively, to create a DVC, one transforms iteratively a starting image $x_0$ to an image $\hat{x}_0$ showing features of the target class $k$ and being close to $x_0$. For this, at every iteration, the unconditionally trained diffusion models provide direction towards the natural distribution of images, the classifier provides direction towards the target class $k$ and the distance regularization keeps the generated image close to the original image $x_0$. We describe this process in more detail below and in S1 Text.

The exact reverse process transition probabilities of a diffusion model guided by an external classifier is intractable. However, similar to unconditional reverse transition probabilities, the conditional ones can be approximated by a Gaussian distribution. This Gaussian distribution has a mean which is shifted from the mean of the unconditional reverse transition probabilities by a term which depends on the gradients of the output of the classifier with respect to the input image. While guidance from the classifier ensures that a sample from the specified target class is produced, it does not guarantee that the generated sample is close to the original image and preserves the retinal structures present in the original image. To address this, we found it beneficial to use a distance regularization term which penalizes the distance between the original image and an approximation of the original image calculated from the current noisy image at any intermediate time step $t$. The overall shift is then a weighted sum of the classifier gradient and the distance regularization terms. For the exact formulation of the distance regularization and the total shift, see S1 Text. To maintain consistent weights across all different images, an adaptive parameterization as discussed in [25] is important. As a further measure to avoid generating images that deviate too much from the original, we start the reverse of the diffusion process from the noisy image at step $\frac{T}{2}$ instead of the completely distorted version of the image at the last step $T$ [25] (Fig 1).

The quality of the generated counterfactuals depended on the type of classifier used for guidance. While adversarially robust models have stronger generative properties, they suffer from a considerable drop in accuracy compared to plain models. As we required a well-performing classifier to guide the counterfactual image generation process to capture clinically meaningful notions of the disease concept, we combined the plain model with better performance while also utilizing the stronger gradients of the robust model. To achieve this, we projected the gradients of the adversarially robust model onto a cone centered around the gradients of the plain model (Fig 1). This procedure is called cone projection [25]. When

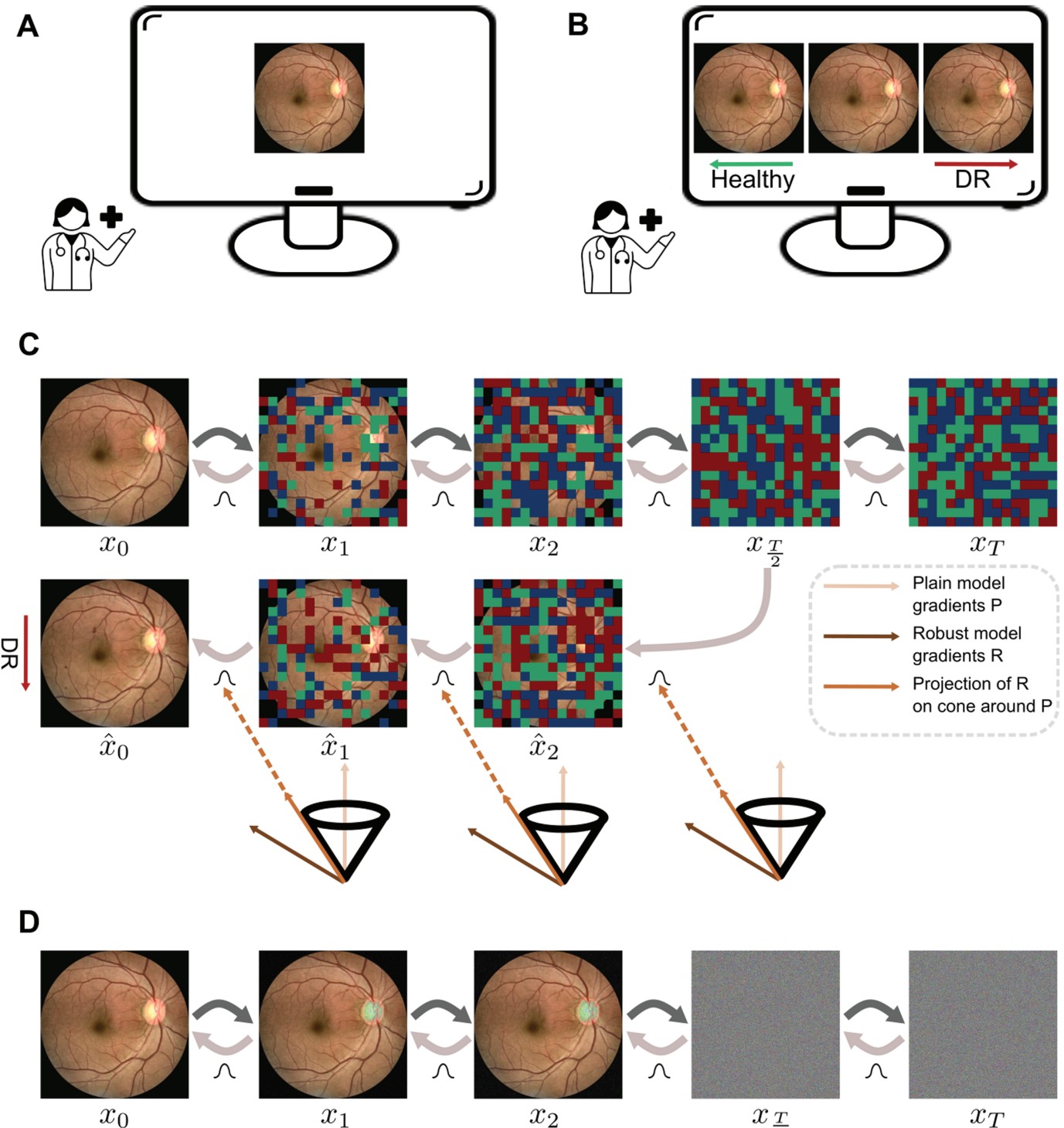

**Fig 1. A. Original retinal fundus image, B. Visualization of counterfactuals with the healthy counterfactual on the left, DR counterfactual on the right and original image in middle, C. Method to generate diffusion counterfactuals.** Top shows the forward and reverse diffusion for an original image $x_0$. Bottom shows generation of a DR DVC starting from the $\frac{T}{2}^{th}$ time step. The mean of distributions in reverse diffusion is shifted using projected gradients (shown in dark orange) of an adversarially robust classifier (shown in brown) on a cone around the gradients of a plain classifier (shown in light orange), D. Images from the actual forward diffusion corresponding to the time steps shown in C. Physician icon from WikiMedia Commons under Creative Commons CC0 license.

cone projection was used, the projected gradient replaces the gradients of individual classifiers in the mean shift (see S1 Text). Computationally, the counterfactual generation process including cone projection required 6484 MiB memory on an NVIDIA A40 GPU for 80 images divided into 16 batches of size 5 each. The average time taken per batch computed across 16 batches was 5.71 minutes and the average time taken per image calculated for 80 images was 1.14 minutes.

Our code is based on https://github.com/valentyn1boreiko/DVCEs and is available at https://github.com/berenslab/retinal_image_counterfactuals.

## Alternative method: Sparse visual counterfactuals

Previous studies on generating retinal counterfactuals either use StyleGANs [8] or adversarially robust classifiers [6]. While the StyleGAN approach is closer to our approach as it uses a generative model, the code or model information is not adequately provided for reproducing the results presented. Hence, for comparison, we used a previously suggested method for generating Sparse Visual Counterfactuals (SVCs) requiring an adversarially robust classifier [5] or at least an ensemble of plain and adversarially robust classifiers [6]. Sparse counterfactuals are computationally similar to adversarial examples but conceptually different from them due to the fact that sparse counterfactuals show meaningful changes that are relevant to the target class instead of the imperceptible noise added to original examples.

The sparsity and degree of realism of the generated counterfactuals can be controlled by changing the norm used for defining the neighborhood of the starting image (S1 Text). A norm of $\ell_4$ was shown to generate the most realistic counterfactuals among the various norms [6]. Since closed-form projections are not possible for the $\ell_4$ norm, the Auto-Frank-Wolfe (AFW) algorithm [5] was used to solve the optimization and generate sparse counterfactuals [6].

The main drawback of sparse counterfactuals is that visual inspection showed that healthy counterfactuals from the DR class using retinal fundus images covered up the lesions on the fundus image with artificial looking blood vessels in a previous study [6]. Although this achieved the effect of removing the lesions to make the image look healthy, these changes do not appear realistic (e.g. see Fig 2, second row in [6]). A more realistic change would have been to cover up the lesions with the background colors instead of adding artificial structures.

## User study

To evaluate the realism of the generated counterfactuals, we performed a user study with trained ophthalmologists as well as AI researchers. We built a web-based image evaluator based on the Python web framework Django (v. 4.2.1) with a PostgreSQL (v. 15.3) back-end database, available at https://github.com/berenslab/retimgtools/tree/v.1.0.0 and showcased in S1 Fig. On the front-end, we used custom JavaScript to modify various presentation parameters (e.g. hiding the images after a certain number of seconds).

Seven ophthalmologists who had a clinical experience of $2, 4, 5, 9, 9, 10$ or $14$ years participated in the study (including author LaK). In addition, 4 AI researchers working on applying deep learning for clinical tasks in ophthalmology took part (including authors PB and LiK).

**Realism assessment method.** All participants were given a three-way odd-one-out task where they had to identify the generated counterfactual among three images. This task design is recommended for this type of study as it is highly sensitive for detecting the odd-one-out

category [30,31]. Each trial thus consisted of two real images from the data sets and one counterfactual generated by a model. Images were displayed for a maximum of 20 seconds and then hidden. All 3 images in any question belonged to the same class. For example, for a question showing DR images, we show two real DR images and one generated counterfactual with DR as the target class. The latter is generated from an image which is labeled as healthy in the data set and classified as healthy by the classifier.

For retinal fundus images, a total of 80 trials were performed with a randomly chosen set of 40 questions showing sparse counterfactuals and the remaining half showing diffusion counterfactuals as the generated image. Within each group, 50% questions belonged to the healthy class and the rest to DR. For OCT scans, on the other hand, only diffusion counterfactuals were shown as the generated images in all 80 questions as study time was a limiting factor with four disease categories. Questions were equally split across the four disease categories with 20 questions for each class. Similar to the fundus scenario, OCT counterfactuals for questions belonging to the healthy category are generated from any of the three disease classes and vice-versa.

**Meaningfulness assessment method.** To test the meaningfulness of the diffusion counterfactuals, we conducted a second user study with five ophthalmologists (with 3, 4, 5, 9, 14 years of experience, including author LaK). We used 20 real images (10 healthy and 10 with referable DR) as well as 20 counterfactuals (10 healthy counterfactuals generated from DR images and 10 images with referable DR generated from healthy images). The task was to grade the images for referable DR. We reasoned that if the changes were not meaningful there would be a difference in grading performance between real and counterfactual images. The images were shown in pseudorandom order.

**Selection criteria.** First, we selected 200 images from the EyePacs test set with 40 images from each class. We generated both DVCs and SVCs for these chosen set of 200 images. Then for both categories of generated images, we included the images for which the classifier correctly classified the original images and had a target class confidence greater than 0.5 on the generated image. The rest of the images were excluded. After filtering, 73 originally healthy and 85 DR images were available for the clinical study. From these, we randomly selected 20 referable DR SVCs and DVCs generated from healthy images and vice versa. This resulted in 80 generated images for the study. 160 real images to be paired with these generated images were sampled from the original set of 200 images such that 40 images belonged to healthy class, 40 to mild class, 40 to moderate class, 20 each to the severe and proliferative classes.

For OCT, we largely followed a similar procedure as for fundus images. Initially, we selected 160 images in total with 40 images from each of the 4 classes. From the 120 images belonging to CNV, drusen and DME class, we generated healthy DVCs. We generated 40 DVCs to each disease class from the set of original healthy images. Here no images were excluded after the filtering step as all the chosen images were correctly classified while also achieving high target class confidences on the generated images. We then randomly selected 20 generated DVCs from each category to form a set of 80 generated images for the questionnaire. All 160 images in the original set were used as real images.

For the meaningfulness evaluation, the generated images chosen were ensured to have a high degree of realism. All the 20 generated images that were chosen for this study were identified as real images by more than 5 participants in the realism assessment. The real DR images were mostly grade 1 (mild) or 2 (moderate) so that the task was quite difficult for clinicians. Three out of the 20 images belonged to the severe and proliferative disease stages.

### Ethics

Ethical approval for the study was obtained from the ethics commission at the University Clinic, Tübingen (Ref No. 250/2023BO2). We obtained consent from all participants in the user study.

### Statistics

Statistical analysis was performed using R (v 4.3.2). A generalized linear model was used to assess the statistical significance of the factors model type (SVC vs. DVC), disease label (healthy vs. DR and normal, CNV, drusen and DME) and expertise level of the grader (ophthalmologist vs. AI researcher) using the *glm* function from the package *stats* (v. 4.3.2). Custom code in Python was used for data visualization.

### Role of funders

The funders did not have any role in in study design, data collection, data analyses, interpretation, or writing of report.

## Results

Our goal was to develop and validate an AI algorithm to generate counterfactuals based on diffusion models guided by the gradients of robust disease classifiers. To this end, we show that the algorithm can generate minimal, meaningful and high-confidence changes to an input image such that the DNN classifier alters its decision to a desired target class and that domain experts view the resulting images as realistic. For fundus images, for example, this means that a healthy fundus image of a given patient is changed into a diseased image by adding disease related lesions while maintaining features such as vessel and optic disc structure related to patient identity, and a diseased image is likewise changed to a healthy one. We validate the realism and meaningfulness of the generated images through an user study with domain experts. We then go into the technical factors necessary for achieving this result, establish that robust classifiers are indeed necessary and illustrate the effect of regularization strength on the generated images. Finally, we demonstrate multi-class counterfactuals and counterfactuals for retinal OCT scans, for which we also evaluate the realism in a user study.

### Fundus diffusion counterfactuals are realistic

We trained binary plain and robust DNN classifiers for DR based on fundus images with high accuracy on a large and diverse fundus image dataset (Tables 1 and 2). As expected, the robust classifier had lower accuracy than the plain one. In addition, we also trained a diffusion model on the same dataset augmented by additional datasets to add more diseased examples. We used the diffusion model with cone projected gradients in order to generate realistic diffusion counterfactuals such that if an image showed signs of DR, the counterfactual could

**Table 2. Evaluation of plain and robust classifiers in terms of standard and balanced accuracy.**

|         | Binary fundus | | 5-class fundus | | OCT | |
|---------|------|----------|-------|---------|-------|----------|
|         | acc. | bal. acc. | acc. | Quad. $\kappa$ | acc. | bal. acc |
| Plain   | 92.39 | 80.67 | 86.65 | 0.67 | 96.35 | 95.87 |
| Robust  | 90.03 | 74.35 | 83.69 | 0.51 | 95.03 | 93.29 |

either remove these signs ("healthy diffusion counterfactual") or reinforce them ("DR diffusion counterfactual"). Likewise, the diffusion counterfactual could either add signs of DR to a healthy original image or strengthen its healthy appearance. Thus, the model was able to generate images that illustrate what the fundus image of a patient might have looked like, had he or she been more or less progressed in their disease (the definition of a counterfactual). We compared the diffusion counterfactual method to the previously published sparse counterfactuals method [6].

We found that the diffusion model generated visually realistic counterfactual fundus images from either DR or healthy starting images (Fig 2). For example, a DR diffusion counterfactual generated from a DR fundus images enhanced the existing lesions and added new lesions resembling microaneurysms or retinal hemorrhages (Fig 2B, right panel, green arrows). Similarly, a DR diffusion counterfactual generated from a healthy image produced diverse lesions including images regions that resembled microaneurysms, hemorrhages , cotton wools spots or exudates (Fig 2E, right panel). Further, structural aspects of the retina including the blood vessels, macula and optic disc were largely preserved on the diffusion counterfactuals of any given subject's fundus image. In comparison, baseline sparse counterfactuals appeared more artificial (left panels in Fig 2B, C, E, F). Sparse counterfactuals introduced artifacts such as waves around lesions in DR counterfactuals (Fig 2B, E) and lines in healthy counterfactuals (Fig 2C, F); for more examples, see S2 Fig.

To assess whether the generated diffusion counterfactuals are realistic, we performed a user study with four AI specialists, who worked with ophthalmological data on a regular basis, and

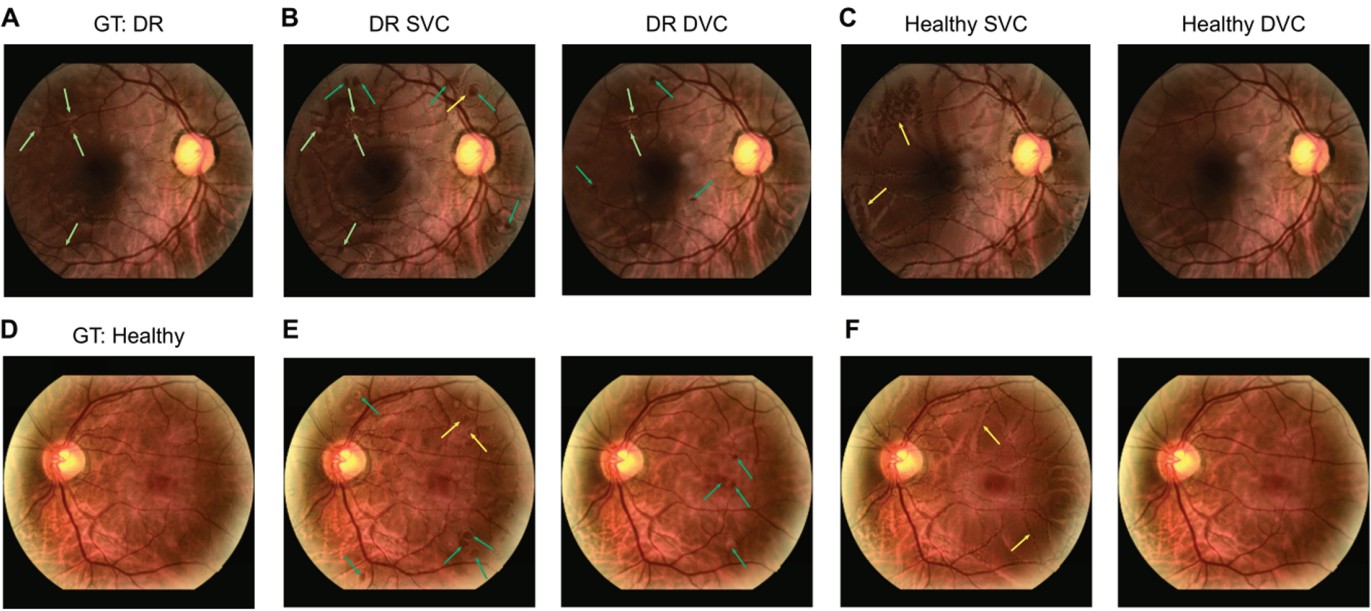

**Fig 2. Diffusion visual counterfactuals (DVCEs) show clinically meaningful changes and appear more realistic than sparse visual counterfactuals (SVCEs).**
A. Color fundus image with signs of diabetic retinopathy (DR). Classifier confidence $p_\phi(\text{DR}) = 0.99$. GT stands for "ground truth", the label assigned to the image in the dataset. B. DR SVC (left) and DVC (right) with $p_\phi(\text{DR}) = 1.00$ for both images. C. Healthy SVC (left) with $p_\phi(\text{healthy}) = 1.00$ and healthy DVC (right) with $p_\phi(\text{healthy}) = 0.99$. DVCs show realistically emphasized lesions (light green arrow) and new lesions (dark green arrow). DVC shows more realistic removal of disease related lesions whereas SVCs introduce artifacts (yellow arrow). **D.-F.**, as **A.-C.**, but for a healthy fundus image $p_\phi(\text{healthy}) = 0.90$. DR SVC: $p_\phi(\text{DR}) = 0.98$; DR DVC: $p_\phi(\text{DR}) = 1.00$; Healthy SVC: $p_\phi(\text{healthy}) = 1.00$; healthy DVC: $p_\phi(\text{healthy}) = 0.99$. All SVCs were generated with $\ell_4$ norm and $\epsilon = 0.3$. DVCs were generated with $\ell_2$ norm and regularization strength $\lambda = 0.5$.

six ophthalmologists with different levels of experience (see Methods). In a three-way odd-one-out task, we asked to identify the image likely to have been generated by an AI model. The shown images included both healthy and DR diffusion and sparse counterfactuals. Interestingly, all participants found it challenging to distinguish diffusion counterfactuals from real fundus images whereas they easily spotted the sparse counterfactuals (Fig 3). In fact, across all images, participants showed a close to chance level (33%) performance for diffusion counterfactuals as opposed to a significantly better performance than chance level for sparse counterfactuals (Fig 3A, DVC vs. SVC: 36.3% [31.7% – 41.1%] correct, 95% CI), confirmed by statistical analysis ($p \ll 0.0001$, see Table 3).

We further analyzed if DR or healthy could be more easily identified as artificial. We found that participants could identify healthy diffusion counterfactuals more easily compared to DR diffusion counterfactuals (Fig 3B), potentially because diffusion models appear to smooth the image during removal of lesions and sometimes fail to remove all traces of lesions ($p = 0.0005$, see Table Table 3). Finally, we studied whether trained ophthalmologists were more likely to identify diffusion counterfactuals than AI specialists. Interestingly, we found that this difference was not major, with all ophthalmologists independent of experience levels being close to chance level, at a similar level as AI specialists (Fig 3C). Ophthalmologists detected sparse counterfactuals at an average rate of 90.4%, significantly better than AI specialists who detected the same at an average rate of 80.6% ($p = 0.0197$, see Table 3).

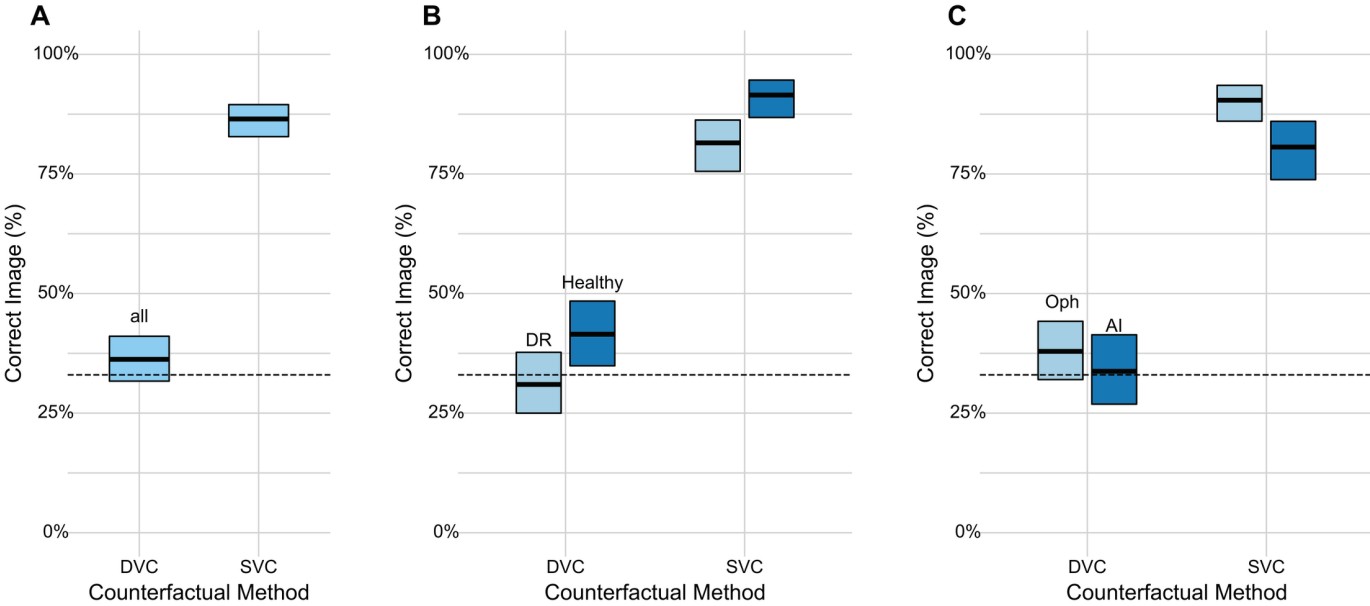

**Fig 3. User study of realism of generated DVCs. We asked** $n = 4$ **AI researchers and** $n = 6$ **ophthalmologists to identify a counterfactual in a odd-one-out task with three images (two real and one counterfactual).** A. Overall fraction of correctly identified counterfactuals with binomial 95%-CI. Baseline at 33% chance level (dashed line). Grey dots and lines indicate individual graders. B. As in A. for the healthy and DR classes. C. As in A. for ophthalmologists and AI researchers.

**Table 3. Generalized Linear Model to assess the influence of factors in Fig 3 n = 800.**

| Predictor | Odds Ratio | CI | p-value |
|---|---|---|---|
| SVC vs. DVC | 12.03 | 8.45 – 17.40 | $\ll 0.0001$ |
| healthy vs. DR | 1.82 | 1.30 – 2.57 | 0.0005 |
| Ophthalmologist vs. AI researcher | 0.66 | 0.47 – 0.94 | 0.0197 |

To assess whether the generated diffusion counterfactuals were meaningful, we performed a second user study, in which five ophthalmologists took part. We reasoned that if changes applied to the fundus images were clinically meaningful, it should not make a difference whether ophthalmologists classified healthy from diseased fundus images based on real images or counterfactual ones (Fig 4A). In contrast, if the applied changes were not clinically meaningful, the grading performance between healthy and diseased images should drop for counterfactual images. To this end, we chose 20 real images (10 healthy and 10 with referable DR) and 20 counterfactual ones (10 healthy images, generated from images originally labeled as referable DR, and 10 images with referable DR, generated from images originally labeled as healthy). Indeed, we found that there was very little difference between the grading performance for the two types of images (Fig 4B, 81% vs 78%, $p = 0.6$, Generalized Linear Model, Odds Ratio for counterfactual vs. real image: $0.83[0.41 – 1.65]$, $n = 200$), indicating that our diffusion counterfactuals contained indeed clinically meaningful changes. Since the DVC method is likely to generate smaller and moderately sized lesions as opposed to larger ones, the 10 generated DR images did not consist of any extremely diseased samples. This directly implies that this task is more challenging for the generated images than the real ones which included a few extreme cases. Despite this, the performance on the generated set is on par with the performance on the real set.

In summary, we found that our diffusion counterfactual model could generate realistic fundus images from both healthy and DR images, emphasizing or removing signs of DR in a clinically meaningful way. We showed that the images generated by our new model are hard to detect even for highly trained experts, in contrast to images created by previous techniques.

## Realistic counterfactual examples require robust classifiers

Now that we established that we were able to generate realistic looking counterfactuals for fundus images, we explored the technical ingredients necessary to achieve this. First, the gradients of the output a standard classifier with respect to the image often do not represent

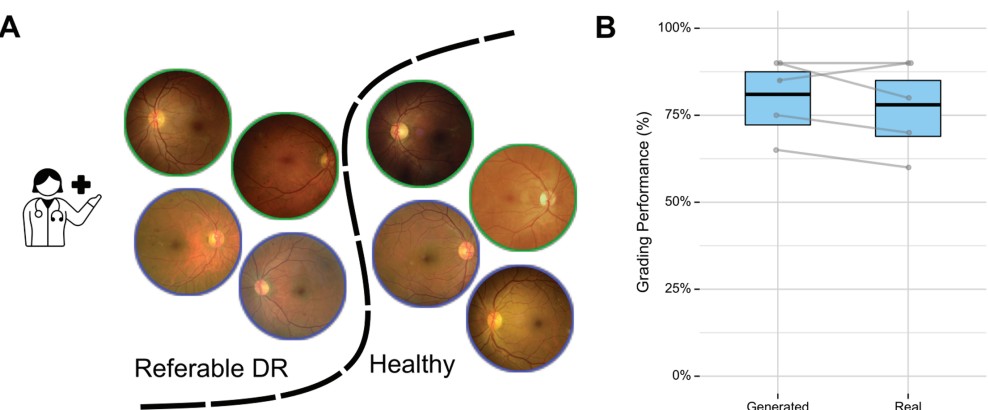

**Fig 4. User study of meaningfulness of generated DVCs.** A. We asked $n = 5$ ophthalmologists to classify a given set of fundus images into healthy and referable DR categories. The image set contained both real fundus images (indicated by green outline) and generated DVCs (blue outline). B. Overall fraction of correctly graded images with binomial 95%-CI for each subset. The performance of clinicians on the generated subset was comparable to that on the real subset showing that the generated DVCs faithfully introduce meaningful features of each class. Grey dots and lines show individual graders. Physician icon from WikiMedia Commons under Creative Commons CC0 license.

meaningful changes, but rather lead to the generation of adversarial examples that fool the classifier but are imperceptible for humans [32]. In fact, for our diffusion model, the gradients of a "plain" classifier often were not strong enough to provide guidance towards the target class and hence, the resulting counterfactuals looked quite similar to the original image (Fig 5A–C, top row). This effect was more prominent in DR diffusion counterfactuals generated from healthy images, where the plain classifier's gradients induced hardly noticeable lesions, compared to healthy counterfactuals generated from DR images, where the diffusion model removed lesions even when guided by the plain classifier (compare Fig 5, top row, to S3 Fig). We investigated the influence of the robust classifier in more detail in S4 Fig.

In contrast, the gradients of the output of a robust classifier with respect to the image supported the generation of high quality DR diffusion counterfactuals, with clearly visible and highly realistic lesions resembling microaneurysms and retinal hemorrhages (Fig 5A–C, middle row). As discussed, the robust classifier, however, traded robustness against accuracy,

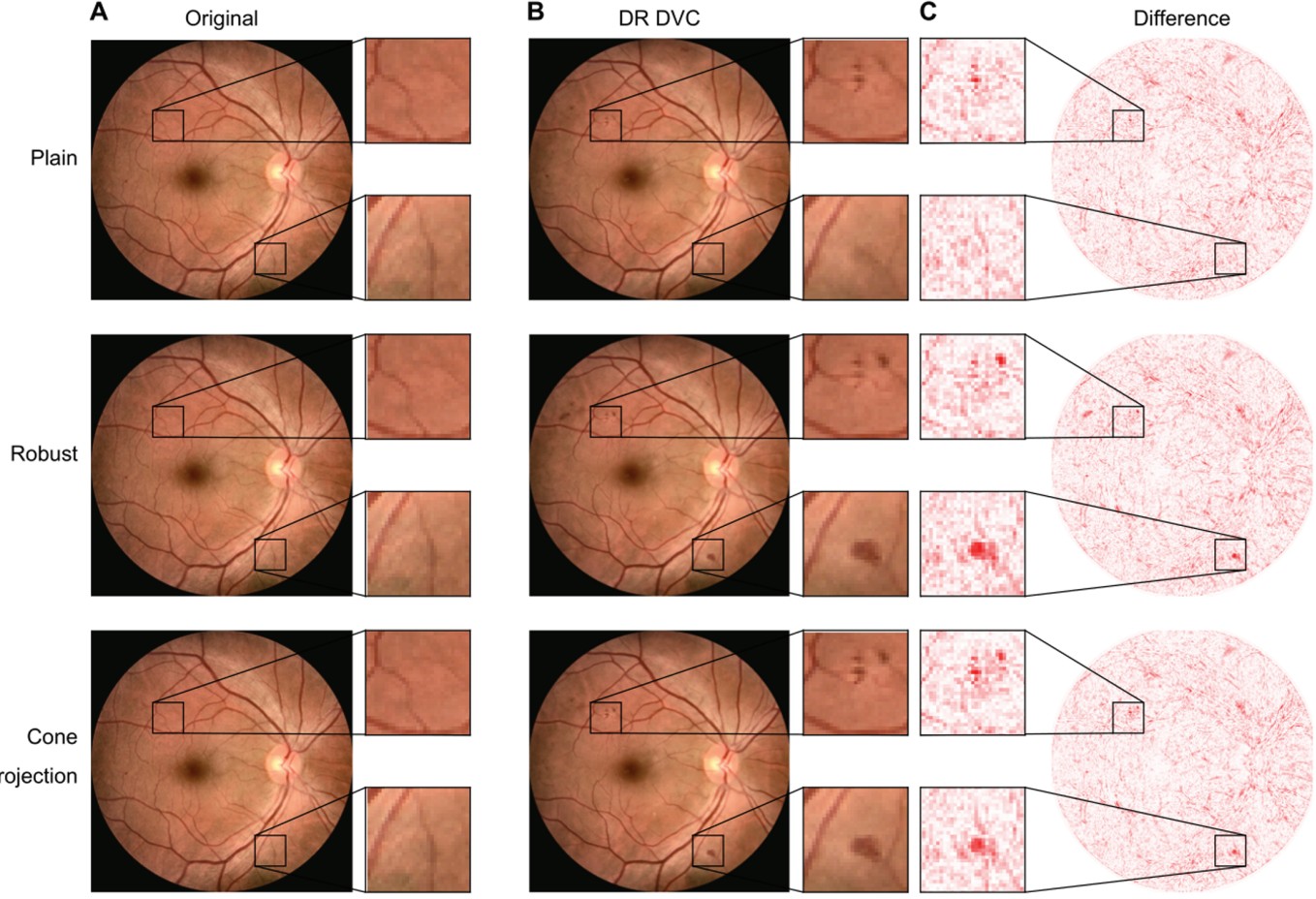

**Fig 5. Comparison of DVCs generated using the plain model (top row), robust model (middle row) and cone projection of an adversarially robust model onto a plain model (bottom row).** A. A DR fundus image with $p_\phi(\text{DR}) = 1.00$ with a zoom in on patches with lesions. B. DR DVCs for the image from A. for the three different models. B. Difference maps between original DR image and the DR DVC show robust and cone projection models produce more realistic changes than the plain model

leading to a drop in performance (see Table 2). To obtain high quality diffusion counterfactuals while maintaining high classification accuracy, we combined the plain and the robust classifier gradients using cone projection. Here, the gradients of the robust classifier are projected onto a cone around the gradients of the plain classifier. In this case, the generated DR diffusion counterfactuals were almost as good for the robust classifier alone (Fig 5A–C, S5 Fig), while maintaining a high balanced accuracy (Table 2). Therefore, our final model evaluated in the user study above used cone projection for generating realistic diffusion counterfactuals. In the sections that follow, all diffusion counterfactuals were generated using the cone projection method.

## Influence of regularisation strength on diffusion counterfactuals

Next, we explored the effect of the regularization strength $\lambda_d$, which constrained the distance of the generated diffusion counterfactual from the original image (S1 Text). This parameter controls the extent of changes appearing on the diffusion counterfactuals compared to the original image, with high values indicating that the generated image remains closer to the original. Without regularization, the diffusion model is not constrained to keep the generated image close to the original image and guided by the gradients from the classifier, it can generate any image belonging to the target class without necessarily preserving the background of the original image including color and vessel structure.

We systematically observed the pattern of changes in both DR and healthy diffusion counterfactuals as we lowered the parameter $\lambda_d$. In general, for both DR and healthy diffusion counterfactuals, more changes were visible on images as $\lambda_d$ was decreased. For DR counterfactuals, the size, number and sharpness of lesions increased with decreasing regularization strength. Thus, with a strong regularization of $\lambda_d = \{0.7, 0.5\}$, fewer lesions appeared on DR diffusion counterfactuals which were relatively smaller and in some cases not too sharp and distinct (Fig 6A). As the strength was decreased to 0.3, more lesions were generated and their sharpness increased compared to the ones generated with higher regularization values. For all these values of $\lambda_d$, almost all counterfactuals were labeled as DR (see Table 4). At 0.2, the diffusion model had the freedom to make several modifications to the original image and it added various bright and large lesions while also modifying the blood vessel structure to a larger extent compared to the other regularization values.

We repeated the above for healthy counterfactuals. Notably, we found that lesions were generally removed well in mild and moderate DR already at larger regularization strengths of $\lambda_d = \{0.7, 0.5\}$, but some traces of lesions were still visible with for severe, proliferative and a few moderate cases (Fig 6B). As regularization was decreased further to 0.3, the sharpness and clarity of those lesions decreased. At $\lambda_d = 0.2$, the lesions were completely removed, however, the blood vessel structure was also heavily altered. Interestingly, also some vessels appeared straighter in the counterfactuals and less spotty, potentially indicative of the algorithm also picking up vessel-related changes in DR [33]. For extreme cases of DR, where the entire retina was affected such as in a few proliferate cases, even a regularization of 0.2 was not sufficient to remove all the lesions (S6 Fig). Furthermore, with $\lambda_d = 0.7$, a third of healthy diffusion counterfactuals generated from DR fundus images did not change the prediction of the DNN classifier to the healthy class. In contrast, for $\lambda_d = 0.5$ and $\lambda_d = 0.3$, 14% and 6% did not convert to healthy class, respectively (Table 4). As we evaluated the classifier at a "referable DR" scenario, a healthy diffusion counterfactual may contain small traces of lesions as seen in mild DR fundus images even when the prediction of the classifier changes to healthy. For more examples of diffusion counterfactuals with varying $\lambda_d$, see S7 Fig and S8 Fig.

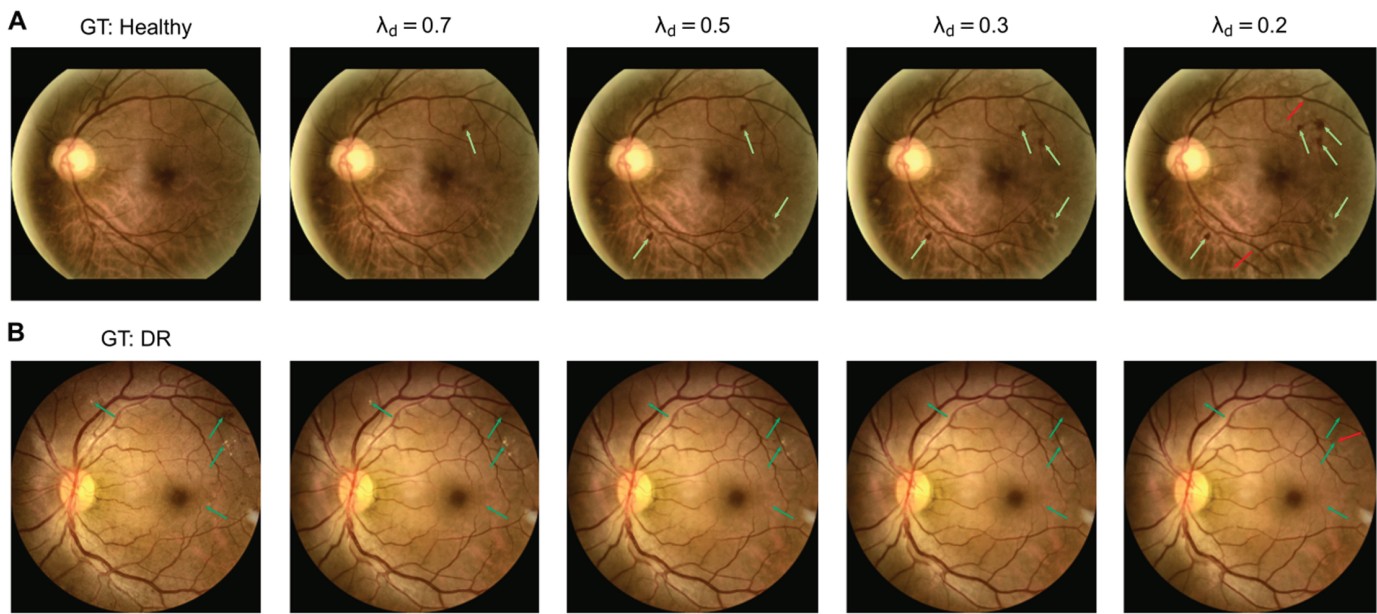

**Fig 6. Effect of tuning the regularization strength $\lambda_d$ on generated DVCs.** Decreasing $\lambda_d$ allows for more changes on the original image. A. We start with a healthy image and generate DR DVCs with decreasing $\lambda_d$. More lesions are generated as $\lambda_d$ decreases (light green arrows). B. We start with a DR image and generate healthy DVCs with different $\lambda_d$ values. Some traces of the lesions were still visible for $\lambda_d = \{0.7, 0.5\}$ while they were completely removed for $\lambda_d = 0.2$ (dark green arrows) at the cost of some changes to the vessel structure (red arrows). While a higher $\lambda_d = 0.7$ is sufficient to generate the minimum number of lesions required to convert a healthy fundus to DR, it is not sufficient to remove all lesions on a DR image to convert it to a healthy fundus.

**Table 4. Fraction of images that do not change the class label depending on the choice of regularization parameter $\lambda_d$.** For this analysis, 40 images were chosen from each of the five classes, such that there were 80 images for the "healthy to DR" direction and 120 for "DR to healthy". For 73/80 and 100/120, class labels were correctly predicted for the original image. Then we evaluated the class label of the corresponding counterfactual.

| $\lambda_d$ | Healthy → DR | DR → healthy |
|---|---|---|
| 0.7 | 4.1% | 27.0% |
| 0.5 | 1.4% | 14.0% |
| 0.3 | 0% | 6.0% |

Taken together, we found that using values of $\lambda_d$ such as 0.2 or lower resulted in larger changes to the original image than necessary for conversion to the target class, producing large changes to the vessel pattern. While high $\lambda_d$ such as 0.7 and higher was sufficient for the DR diffusion counterfactuals to show minimal features required to convert healthy fundus to DR, the same did not hold for healthy diffusion counterfactuals generated from DR fundus images (Table 4). Therefore, we chose a regularization value of 0.5 where we could qualitatively observe the minimal changes on the image necessary to alter the decision and confidences of the classifier in both directions while maintaining image structure close to the original (although within a range of $\lambda_d = 0.3 - 0.5$, this is a qualitative judgment).

### Diffusion counterfactuals for the multiclass DR grading task

We followed up the counterfactuals for the binary case of healthy versus DR with counterfactuals for a more fine-grained classification scenario with 5 classes, healthy, mild, moderate, severe and proliferative. The latter four categories are the various stages of DR in the order of

increasing severity. The mild class often shows only very tiny changes in the form of microaneurysms and is the hardest to detect. Moderate and severe are characterized by the presence of a relatively greater number of microaneurysms and larger lesions such as hemorrhages and exudates. The proliferative class is the most advanced stage with venous bleeding, large hemorrhages and neovascularization. Some of the images in the proliferative and severe stages also show scars resulting from laser treatment. Typically severe and proliferative classes are easier to detect due to larger lesions however due to rare occurrences they are underrepresented in the data set.

We generated diffusion counterfactuals to the 5 different classes from originally healthy, mild and moderate fundus images (Fig 7). First, we looked at diffusion counterfactuals to the various DR stages from a healthy image and found that the diffusion counterfactuals contained meaningful features for both mild and moderate classes. The features included tiny dot-like microaneurysms/exudates for mild class and slightly larger and more exudates,

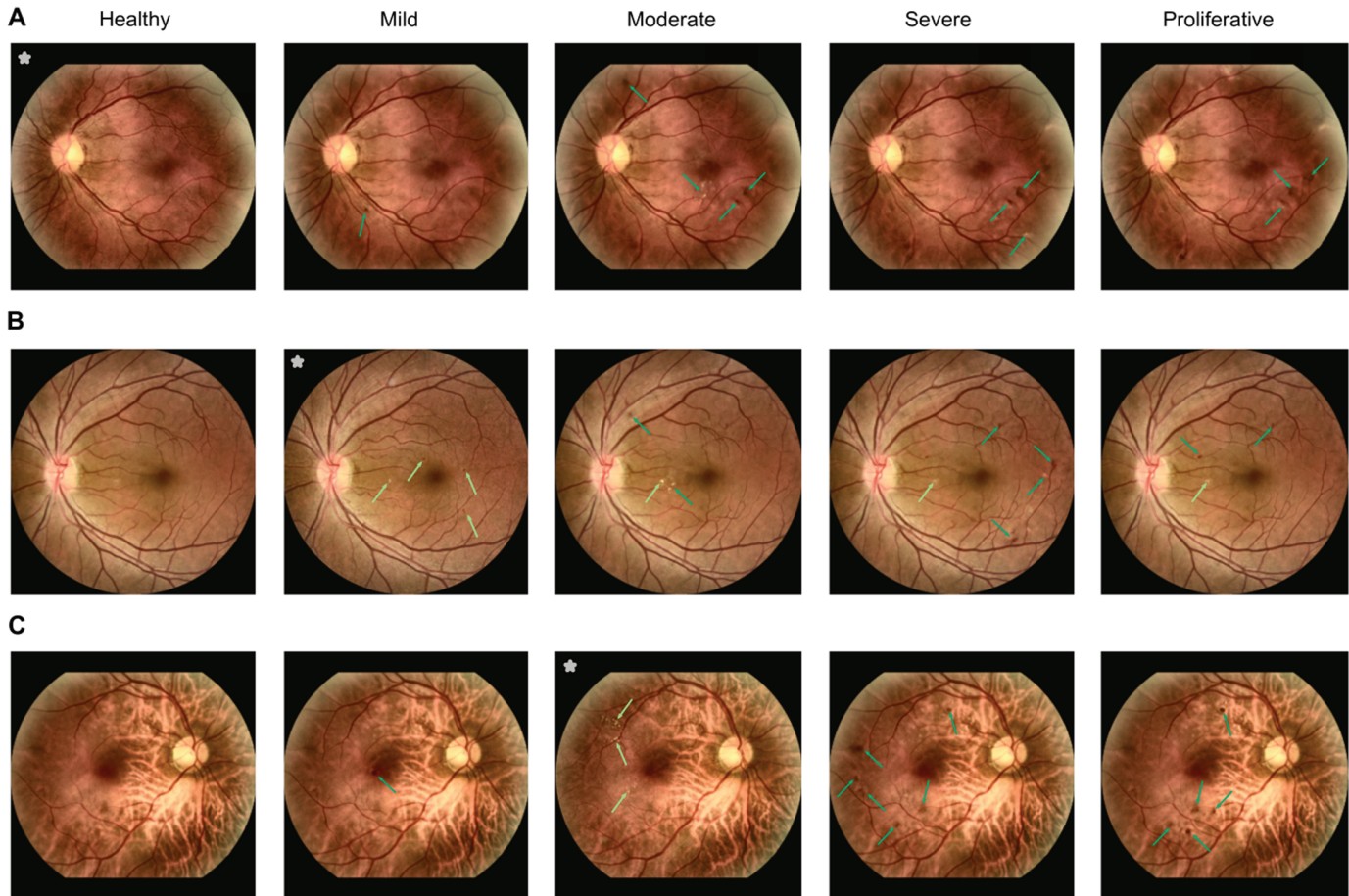

**Fig 7. DVCs for DR grading task with 5-classes: healthy, mild, moderate, severe and proliferative.** Images marked with * are original images with GT as indicated in the headline. All other images are DVCs with the headline specifying the target class. DVCs to the different classes from a A. healthy fundus, B. fundus with mild DR and C. fundus with moderate DR. Lesions which are originally present in initial image are indicated with light green arrows while lesions added by DVC are indicated with dark green arrows. In all cases, healthy DVCs removed all lesions. While the number and types of lesions introduced in mild and moderate DVCs are consistent with those observed in real-world data, severe and proliferative DVCs did not reflect the size and intensity of lesions seen in real examples. The model fails to generate larger lesions as seen in severe and proliferative classes due to the low representation of these classes in the data set. Other failure cases can be seen in S9 Fig.

microaneurysms and a few haemmorhages for moderate class (Fig 7A). However, for the severe and proliferative classes most often only a couple of scattered haemmorhages were generated and most other features such as bleeding or the laser scars were not observed (Fig 7A). This was likely due to the scarcity of these classes in the data set. Another technical factor could be the choice of parameters such as the regularization value $\lambda_d$ and the radius $\varepsilon$ which were more suitable for smaller changes.

For healthy diffusion counterfactuals from both mild and moderate classes, all lesions were removed completely in most cases with a regularization strength of 0.5 (Fig 7B, C). On a moderate diffusion counterfactual generated from a fundus image originally belonging to the mild class, the number of exudates increased and an existing exudate was slightly enlarged (Fig 7B). The diffusion counterfactual from moderate to mild interestingly removed the exudates all over the fundus and added a single microaneurysm (Fig 7C). Here again, diffusion counterfactuals to the more advanced stages of severe and proliferative did not exhibit the relevant features for those classes (Fig 7B, C). Furthermore, in all cases, the plain classifier achieved high target probabilities in the range [0.85, 1.00] for diffusion counterfactuals to the healthy, mild and moderate classes while having only low target probabilities which dropped to below 0.25 for the severe and proliferative diffusion counterfactuals.

To summarize, the 5-class model could generate meaningful diffusion counterfactuals to the healthy, mild and moderate classes while it was not as efficient at generating severe and proliferative cases. Nonetheless, mild and moderate classes are the clinically more interesting stages as they are challenging to detect and diagnostic decisions are uncertain for not only DNNs but also ophthalmologists around the boundaries of these early stages [34]. Studying the progression of biomarkers closely in these stages with counterfactuals can help prevent conversion to the more advanced stages.

## Diffusion counterfactuals of OCT scans are also realistic

Finally, we trained another set of diffusion model and classifiers on a database of OCT B-scans, a different image modality which is also predominantly used in ophthalmology (Table 1). The diffusion model was trained to generate realistic OCT scans and the task of the classifiers was to detect whether a given scan was healthy or had one among the three conditions: choroidal neovascularization (CNV), drusen or Diabetic Macular Edema (DME), which both plain and robust classifiers were able to do with high accuracy (see Table 2).

OCT B-scans visualize a cross-section of the retina and typically show its different layers, from the vitreo-retinal interface, inner retina, outer retina, retinal pigment epithelium (RPE)/Bruch's membrane to the choroid from top to bottom. The biomarkers of CNV on OCT scans included subretinal neovascular membrane, subretinal fluid and intra-retinal fluid. These occur due to abnormal growth of new vessels in the choroid creating a rupture in the retinal layers above the Bruch's membrane. Drusen were characterized by a bumpy or irregular RPE layer due to lumps of deposits under the RPE. OCT scans of subjects with DME contained several cavity-like structures in the inner and sub retinal layers which represent intraretinal and subretinal fluid that accumulates due to vascular leakage [35].

We generated diffusion counterfactuals using the cone projection method with the OCT scans to the three disease categories from healthy and vice-versa. Qualitatively, diffusion counterfactuals contained meaningful changes matching the clinical features described above. For example, the diffusion counterfactual to CNV from healthy added subretinal fluid below the RPE (Fig 8A, top row), and the diffusion counterfactual from CNV to healthy removed the subretinal neovascular membrane and flattened out the portion where it was present (Fig 8A, bottom row). Also, diffusion counterfactuals from healthy to the drusen class showed

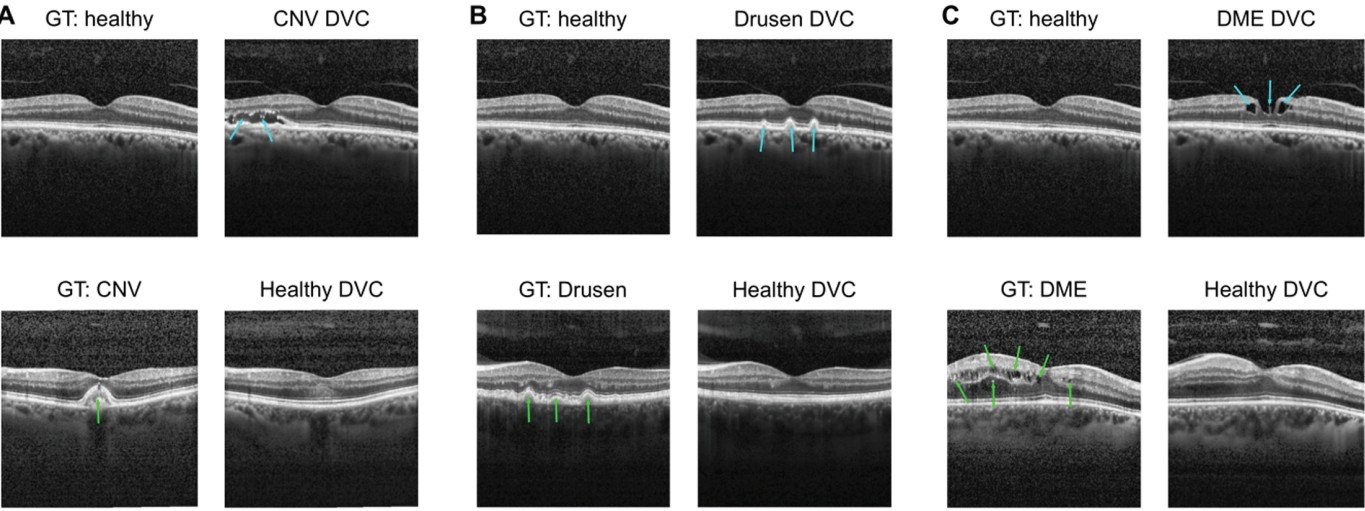

**Fig 8. DVCs for B-scans from optical coherence tomography (OCT) from healthy to various disease classes and vice-versa.** A. DVC from healthy to choroidal neovascularization (CNV) (top) and from CNV to healthy (bottom). **B,C.** Same as **B** for classes drusen (**N**) and diabetic macular edema (DME) (**c**). Similar to fundus DVCs, OCT DVCs show meaningful changes which are consistent with the important features of each class. DVCs from healthy images add features relevant to the disease (blue arrows). DVCs from diseased images to the healthy class remove the disease specific features seen on original image (green arrows). Upon visual inspection, OCT DVCs are more realistic than SVCs. For SVCs of the above OCT images, see S10 Fig.

added bumps to the RPE layer (Fig 8B, top row); in the reverse case the irregularities were removed to make the RPE layer smooth and flat (Fig 8B, bottom row). DME diffusion counterfactuals generated from the healthy class contained cavities in the inner retinal layers (Fig 8C, top row) and healthy diffusion counterfactual from a DME OCT scan covered up the cavities with the original tissue reflectivity (color) in those layers (Fig 8C, bottom row).

To quantitatively assess the degree of realism of the generated images, we again performed a user study with six ophthalmologists and four AI researchers. Similar to the fundus user study, the participants were assigned a three-way odd one out task here too although they were shown only diffusion counterfactuals across the four different categories. Ophthalmologists consistently performed better than chance (33.3%) in all classes (Fig 9, indicated by non-overlapping 95%-CIs). Interestingly, they detected diffusion counterfactuals to the normal class from the various disease classes with the highest rate of 66.7%. This could have been due to the normal diffusion counterfactuals generated from OCT scans with signs of extreme CNV or DME. In such scenarios, the normal DVCEs generally tended to fill up the cavities or tears with original tissue reflectivities but did not restore the thickness of the layers (Fig 8C, bottom row) thereby resulting in easier detection. They found diffusion counterfactuals to the CNV and drusen classes also easier to detect (Fig 9). This could be due to certain features that looked artificially generated for e.g. the perfect waves on drusen diffusion counterfactuals. On the other hand, diffusion counterfactuals to the DME class were the hardest to detect for the ophthalmologists (Fig 9). AI researchers performed significantly worse overall (see Table 5), in particular for CNV and DME (Fig 9), likely due to the lack of experience of AI researchers with these disease categories. Taken together, OCT diffusion counterfactuals were able to generate the primary features associated with each class convincingly although with a few imperfections which led to their easier detection in the user study compared to fundus diffusion counterfactuals.

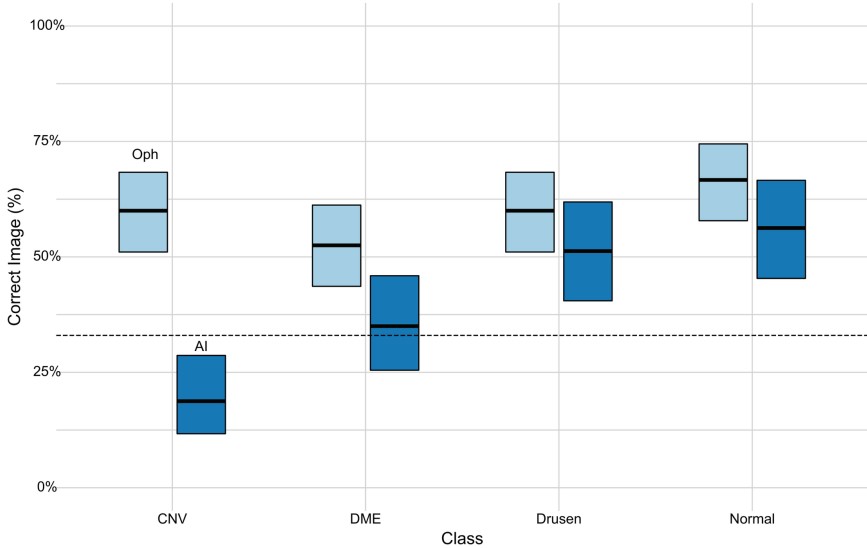

**Fig 9. Clinical evaluation of realism of generated OCT DVCs. We asked** $n = 4$ **AI researchers and** $n = 6$ **ophthalmologists to identify a DVC in a odd-one-out task with three images (two real and once DVC).** A. Overall fraction of correctly identified DVCs with binomial 95%-CI. Baseline at 33% (dashed line). B. As in A. OCT DVCs are easier to detected by ophthalmologists mainly because the changes are to and from more progressed disease stages, which require more attention to the global image structure, compared with fundus DVCs.

**Table 5. Generalized Linear Model to assess the influence of factors in Fig 9 n = 800.**

| Predictor | Odds Ratio | CI | p-value |
|---|---|---|---|
| CNV vs. DME | 1.09 | 0.73 – 1.63 | 0.6817 |
| CNV vs. drusen | 1.72 | 1.15 – 2.58 | 0.0082 |
| CNV vs. normal | 2.23 | 1.49 – 3.37 | 0.0001 |
| Ophthalmologist vs. AI researcher | 0.44 | 0.33 – 0.60 | < 0.0001 |

## Discussion

### Summary

Realistic counterfactuals can help to provide human-like and human-understandable reasoning for image-based deep learning models [3]. We rigorously evaluated classifier-guided diffusion models for generating counterfactual images for retinal imaging filling an important gap in the literature [36]. In a user study, we found that these images could hardly distinguish from real images by domain experts in ophthalmology and they contained clinically meaningful information, opening up new opportunities to include counterfactual images in medical reasoning or training [2,3].

### Potential applications

Such counterfactual images could in principle serve as a powerful visual tool for patient education, illustrating potential outcomes based on different clinical decisions or behavioral patterns. For instance, in patients with diabetic retinopathy (DR), counterfactual images could be generated from their current fundus image to highlight the phenotypical consequences of good or poor glycemic control. Given that DR manifests through visibly evident

fundus changes, these scenarios could provide striking and highly illustrative examples for patients, potentially enhancing patient motivation and adherence by making the impact of their choices more tangible.

Alternatively, counterfactual images could aid training and education of future medical professionals, where generating realistic counterfactual images offers a valuable resource, especially when case examples are scarce or difficult to locate within internal databases. These images can present coherent disease progression scenarios, illustrating the evolution of lesions and other pathologies over time. Synthetic data generated in this way can enrich training datasets, creating more comprehensive and robust educational setups.

In principle, counterfactual images could also be used for decision support and for resolving uncertainty in the diagnosis, where one could generate DVCs to show how the imaging data might have looked like if it had provided a more certain evidence for the presence of a disease. The clinician could then use similarity of the present image to the generated counterfactuals to judge the presence or absence of disease signs. However, this application is made difficult by the fast pace of clinical practice as compared to the time needed for image generation, which might be overcome in the future.

Other applications address more the training of medical decision systems. As medical data sets are often imbalanced and diseased samples may be less readily available, more diseased examples could be synthesized using our method based on a preliminary classifier from the more prevalent healthy examples in the data set [37,38]. This additionally synthesized data could aid in augmenting DNN models. Adding a diseased counterfactual for each healthy image and vice versa would effectively create a paired dataset, where structurally similar images derived from the same base image are contained in the healthy and diseased class, allowing models to focus more easily on the disease patterns. In the reverse case of generating healthy examples from diseased, the counterfactuals could help in anomaly detection and identification of bio-markers [11,12,39]. Counterfactual images could be used for auditing medical AI systems to ensure that the classifiers do not use any shortcuts to make the decisions, such as hospital or device logos instead of disease related features [3,6,40].

## Limitations

While the counterfactuals for retinal fundus images were nearly indistinguishable from real ones for clinicians, OCT counterfactuals were easier to detect. Likely, one factor is that lesions for early DR stages visible in fundus images are mostly localized and not too large, therefore not requiring major structural changes to large parts of the image, in contrast to what is needed for generating OCT counterfactuals. Also, OCT counterfactuals typically looked too regular and symmetric, such as in the case of drusen counterfactuals. Qualitative feedback after the user study indicated that raters were quickly able to pick up on these regularities. In addition, healthy OCT counterfactuals from extremely diseased cases often covered the abnormalities with appropriate texture but did not alter the thickness of the retina which is an important factor for clinicians to classify the image as healthy. Since most examples in the chosen OCT data set also belong to extreme disease stages, this could have further impacted the overall performance of clinicians in the detection of OCT counterfactuals. In contrast, fundus images covered the whole disease spectrum, allowing the model to learn about the gradual changes along the disease trajectory.

It is possible that the different degrees of realism of fundus vs. OCT counterfactuals come from the generative capabilities of the diffusion model. In fact, we noticed that supplementing

the fundus image dataset with more examples from diseased classes helped to generate better disease counterfactuals overall, although we were not successful in adding sufficiently many example of severe and proliferative DR such that model was able to generate realistic counterfactuals to these classes. It is also possible that the disease concepts learned by the classifiers which guided the diffusion model were insufficient. For example, as the DR dataset only had comparatively few samples with media opacity and vitreous hemorrhages in advanced DR classes and the variability in these classes was high, such features were never generated on the DVCs, even after adding additional datasets, potentially because the classifiers did not use these features for making a decision. Instead, the classifiers had seen an abundance of DR images with microaneurysms, hemorrhages and exudates and therefore tended to add these features when queried for a DR DVC. While we did see some alterations in the structure of small vessels in the healthy counterfactual images that could be related to the removal of DR-related changes, these were not very prominent, likely reflecting also the fact that vessel related changes are only a diagnostic criterion for DR in severe stages not captured well by out model [33].

Similarly, with the OCT dataset consisting mostly of extreme examples at advanced disease stages and missing much of the borderline cases in between [24], a classifier might have easily taken a short-cut towards distinguishing healthy from diseased images, picking up on the most informative feature (e.g. texture) while ignoring more subtle or difficult to model ones (retinal thickness) [41]. More broadly, the gradient of the classifier used to guide the diffusion process may not always introduce the most important perceptually relevant features and to some extent, the exact nature of features generated may also depend on the type of classifier, its training settings or the choices for making the classifier adversarially robust [13,25]. Therefore, while our method works well in practice, it does not give formal guarantees that a generated image comes from the distribution of the target class.

Finally, it is possible that the properties of the datasets used for training such as the devices used to collect fundus or OCT images and subject demography might introduce biases to the generative process. Due to this, the generation process might not produce the desired results when exposed to data which comes from a distribution different from the training distribution.

## Related work

Diffusion models have been largely used in combination with plain classifiers for realistic counterfactual generation in the natural image domain using data sets such as ImageNet and CelebA [4,36,42]. The quality and realistic nature of counterfactuals for such images has been shown to improve when adversarially robust classifiers are used [25]. In the medical setting, diffusion models have been used primarily for generating healthy counterfactuals [11,12,39], which is an easier task for the diffusion model compared to generating disease related features. We demonstrated that both diffusion models and adversarially robust classifiers play a major role in generating realistic medical counterfactuals for high-resolution retinal fundus and OCT images. Moreover, our counterfactuals are bi-directional, i.e. from healthy to diseased and diseased to healthy. In parallel work, BioMedJourney [43] used two consecutive Chest-XRay images of a subject and a summary of their medical reports to generate longitudinal counterfactuals. Here, a latent diffusion model was trained on embeddings of the textual descriptions and a starting image to obtain an estimate of the progressing image. This method relies on the availability of detailed medical reports in addition to longitudinal imaging data. Another parallel study generated OCT scans with enhanced choroidal visibility using generative adversarial networks [44].

### Future directions

A natural extension of this work would be to generate counterfactuals from multi-task DNNs which learn several attributes simultaneously [45]. With such a DNN, it would be possible to generate counterfactuals for one attribute keeping another fixed. For instance, a multi-task classifier which is trained on both age and disease type can be used to generate counterfactuals for increasing age keeping the disease fixed or vice-versa. Such counterfactuals could potentially be used in tracking the progression of a disease with age. Alternatively, one could use other forms of diffusion guidance like language for targeted image generation. Also, it would be interesting to study counterfactuals for longitudinal data or data with interventions, such as the administration of a drug. For example for OCT images during age-related macular degeneration, treatment effects for the injection of anti-VGEF drugs might be simulated for the different available drugs, and the most promising drug chosen.

Moreover, improved adversarially-robust classifiers with perceptually well-aligned gradients [46] based on more realistic and varied datasets may yield even better counterfactuals. In fact, it would be interesting to study how the generative capacities of the model change with even larger datasets, such as those used to train retinal foundation models, or based on robust classifiers derived from such foundation models [47]. In fact, a potential direction for future research includes combining out method with techniques to remove biases, spurious features and systematic errors [18,48–51] of classifiers. A related practical challenge in real-world implementation of the DVC method is in continuously updating the models with the influx of new data which results in a distribution shift. For the method to function smoothly for the new data, all the component models including the diffusion model, plain and adversarially robust classifiers need to be fine-tuned. Finally, on the methodological side, recent work has shown potential to generate counterfactuals for explaining both classifiers and foundation models without the need to train an adversarially robust model, which has not yet been investigated in the context of medical images [40].

## Supporting information

**S1 Text. Additional explanations, derivations and equations and explanations for diffusion models and diffusion visual counterfactuals.**
(PDF)

**S1 Table. Performance of plain and robust fundus binary and 5-class classifiers under different settings.** Top 2 rows shows evaluation of plain binary classifiers trained with and without including the mild class. The rows below corresponding to Robust shows the evaluation of robust binary classifiers that are trained with different $\epsilon$ values and initializations - ImageNet indicates that model is finetuned from ImageNet pre-trained plain model and Madry Robust indicates that model is finetuned from an adversarially robust model trained on the ImageNet dataset. Bottom 3 rows shows performance of 5-class models using different $\epsilon$ values.
(PDF)

**S1 Fig. Web interface for evaluating realism of counterfactuals.** Three images are shown on the page where two are real and one is generated. User is asked to select the generated image.
(TIF)

**S2 Fig. More examples comparing SVCs and DVCs.** Top two rows show counterfactuals from DR fundus images. Bottom two rows show counterfactuals from healthy images. In all cases, changes in DVCs are more realistic compared to SVCs.
(TIF)

**S3 Fig. As in Fig 5 for a DR fundus image. a**. Original image with ground truth label DR. **b**. DVC to the healthy class. Plain model (top row) removes lesions to a similar extent as robust (middle row) and cone projection (bottom row). DVCs to healthy class are more easily generated than to DR class. **c** Difference maps between the original DR image and generated healthy counterfactual highlighting lesion locations.
(TIF)

**S4 Fig. Effect of adjusting the angle of the cone around the plain classifier gradients on which the robust classifier gradients are projected a–b.** DR DVCs of healthy fundus images with different values of $\alpha \in \{1, 5, 15, 30, 40, 50\}$. Lower angles correspond to a larger weight of plain classifier gradients. With low values $\alpha \in \{1, 5, 15\}$ disease signs are not seen on the DVCs. On the other hand, setting $\alpha$ higher than 30 results in the introduction of meaningful lesions on the originally healthy images implying that the robust classifier's gradients play a significant role in generating DR DVCs. **c–d** Healthy DVCs of DR fundus images with same values of $\alpha$ as in **a–b**. In this case, lesions are effectively removed to generate healthy DVCs even with a lower angle. Plain classifier alone is able to generate a healthy DVC effectively from an originally DR image. This is in line with our findings derived from Fig 5 and S3 Fig and summarized in Sect Realistic counterfactual examples require robust classifiers.
(TIF)

**S5 Fig. More examples of DR DVCs generated from healthy fundus images (leftmost column) using the plain model gradients (second column), robust model gradients (third column) and cone projected gradients (rightmost column).** In all examples, plain models either show no or fewer and weaker lesions compared to robust and cone projection models.
(TIF)

**S6 Fig. Effect of regularization strength on retinal fundus images severely affected by DR.** In such extreme cases, even a small regularization of 0.2 is not sufficient to convert the image to healthy.
(TIF)

**S7 Fig. Effect of regularization strength on retinal fundus images belonging to DR and healthy class. a–b**. Healthy DVCs of DR fundus images with different values of $\lambda_d$. With $\lambda_d =$ 0.5, examples are converted to the healthy class with either no lesions (**a**) or very few remaining lesions (**b**) such as in the mild DR class. **c–d** DR DVCs of healthy fundus images with varying $\lambda_d$. Here too, with $\lambda_d =$ 0.5, the DVC adds enough lesions to change the decision of the classifier to the DR class with high confidence.
(TIF)

**S8 Fig. Effect of higher values of regularization strength $\lambda_d$ on retinal fundus images belonging to DR and healthy class. a–b**. Healthy DVCs of DR fundus images with different values of $\lambda_d \in \{0.9, 1.1, 1.3\}$ . With such high $\lambda_d$ values, DR lesions are not removed and the resulting DVCs do not appear to be healthy **c–d** DR DVCs of healthy fundus images with same higher values of $\lambda_d$ as in **a–b**. Here too, the lesions added are too subtle and hard to notice, thus achieving a low confidence for the DR class.
(TIF)

**S9 Fig. Examples of failure cases of generated DVCs a.** DVC of a healthy image to the healthy class only brightens an artifact present in the image but is still predicted as healthy with high confidence. **b**. DR DVC of a healthy example does not exhibit any disease related

lesions. **c–d**. Healthy DVCs of DR examples still show disease related lesions. In **c**, one of lesions appears brighter in the DVC.
(TIF)

**S10 Fig. For the same original images as in Fig 8, SVCs of OCT images from healthy to various disease classes and vice-versa A.** SVC from healthy to CNV (top) and from CNV to healthy (bottom). **b–c.** Same as **a** for classes drusen (**b**) and DME (**c**). While SVCs to the diseased classes introduce the respective disease signs, they also produce several artifacts on the background. On the other hand, the method struggles to convert the diseased classes back to healthy especially when the signs of disease are extreme as in the CNV case (**a**). For drusen and DME the SVC attempts to cover the disease signs although in an unrealistic manner.
(TIF)

## Acknowledgments

We thank our clinical colleagues who supported the user study with their expertise. David Merle provided input and discussion on potential application scenarios.

## Author contributions

**Conceptualization:** Indu Ilanchezian, Valentyn Boreiko, Murat Seckin Ayhan, Matthias Hein, Lisa Koch, Philipp Berens.

**Data curation:** Indu Ilanchezian, Ziwei Huang, Lisa Koch.

**Formal analysis:** Indu Ilanchezian.

**Funding acquisition:** Matthias Hein, Philipp Berens.

**Investigation:** Indu Ilanchezian, Valentyn Boreiko.

**Methodology:** Indu Ilanchezian, Valentyn Boreiko.

**Project administration:** Matthias Hein, Lisa Koch, Philipp Berens.

**Software:** Ziwei Huang, Murat Seckin Ayhan.

**Supervision:** Matthias Hein, Lisa Koch, Philipp Berens.

**Validation:** Laura Kühlewein.

**Writing – original draft:** Indu Ilanchezian, Lisa Koch, Philipp Berens.

**Writing – review & editing:** Valentyn Boreiko, Laura Kühlewein, Murat Seckin Ayhan, Matthias Hein.

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
