## [Decision Letter · Decision Letter 0]

19 Feb 2025

PDIG-D-25-00003Development and validation of an AI algorithm to generate realistic and meaningful counterfactuals for retinal imaging based on diffusion modelsPLOS Digital Health Dear Dr. Berens, Thank you for submitting your manuscript to PLOS Digital Health. After careful consideration, we feel that it has merit but does not fully meet PLOS Digital Health's publication criteria as it currently stands. Therefore, we invite you to submit a revised version of the manuscript that addresses the points raised during the review process. Please submit your revised manuscript within 60 days Apr 20 2025 11:59PM. If you will need more time than this to complete your revisions, please reply to this message or contact the journal office at digitalhealth@plos.org. Please include the following items when submitting your revised manuscript:* A rebuttal letter that responds to each point raised by the editor and reviewer(s). You should upload this letter as a separate file labeled 'Response to Reviewers'. This file does not need to include responses to any formatting updates and technical items listed in the 'Journal Requirements' section below.* A marked-up copy of your manuscript that highlights changes made to the original version. You should upload this as a separate file labeled 'Revised Manuscript with Track Changes'.* An unmarked version of your revised paper without tracked changes. You should upload this as a separate file labeled 'Manuscript'. If you would like to make changes to your financial disclosure, competing interests statement, or data availability statement, please make these updates within the submission form at the time of resubmission. Guidelines for resubmitting your figure files are available below the reviewer comments at the end of this letter. We look forward to receiving your revised manuscript. Kind regards, Janna HastingsGuest EditorPLOS Digital Health Janna HastingsGuest EditorPLOS Digital Health Leo Anthony CeliEditor-in-ChiefPLOS Digital Healthorcid.org/0000-0001-6712-6626 **Journal Requirements:**

1. We ask that a manuscript source file is provided at Revision. Please upload your manuscript file as a .doc, .docx, .rtf or .tex.

 **Additional Editor Comments (if provided):** The manuscript tackles an important and challenging area of innovation for medical image-based diagnosis in digital health: the generation of counterfactual images for a given patient. The reviewers have made some substantial suggestions for revisions and improvements of the work, which you will find below.**Reviewers' Comments:** Reviewer's Responses to Questions

**Comments to the Author**

1. Does this manuscript meet PLOS Digital Health’s publication criteria? Is the manuscript technically sound, and do the data support the conclusions? The manuscript must describe methodologically and ethically rigorous research with conclusions that are appropriately drawn based on the data presented.

Reviewer #1: Yes

Reviewer #2: Yes

Reviewer #3: Yes

2. Has the statistical analysis been performed appropriately and rigorously?

Reviewer #1: Yes

Reviewer #2: Yes

Reviewer #3: Yes

3. Have the authors made all data underlying the findings in their manuscript fully available (please refer to the Data Availability Statement at the start of the manuscript PDF file)?

Reviewer #1: Yes

Reviewer #2: No

Reviewer #3: Yes

4. Is the manuscript presented in an intelligible fashion and written in standard English?

Reviewer #1: Yes

Reviewer #2: Yes

Reviewer #3: Yes

5. Review Comments to the Author

Reviewer #1: This manuscript presents a novel approach to generate realistic and clinically meaningful retinal fundus images using diffusion models combined with classifier guidance, bridging advanced computer vision methods with domain-specific studies in medical imaging. The authors demonstrate that their model can generate high quality counterfactuals for both CFP and OCT images, validated by technical and clinical experts. The work is technically sound, with a clear explanation of the methods, open-source code and dataset sources, and rigorous evaluation results.

The main contribution of this work is the technical innovation of computer vision methods (generating counterfactuals using diffusion models guided by classifiers) in the clinical domain. The combination of diffusion models, adversarially robust classifiers and cone projection is well motivated and effectively implemented. What's more, expert evaluation by AI specialists and ophthalmologists lends credibility to the results. The three-way odd-one-out task and clinical relevance assessment provide strong evidence that the generated images are realistic and clinically meaningful.

The authors also evaluate the technical improvements over the baseline and analyse the effect of regularisation strength on the generated images. Supplementary materials with additional visual examples and mathematical details increase the transparency and reproducibility of the work.

The manuscript is well written and logically structured, with clear explanations of methods and results. The use of many visual examples throughout the manuscript helps to effectively illustrate the key points.

However, there are some areas for improvement. Firstly, while the authors discuss the role of robust classifiers in generating high quality counterfactuals, a more systematic and quantified ablation study comparing different classifiers or classifier settings could improve technical soundness and provide deeper insights into the behaviour of AI models in clinical terms.

Another area for improvement is that, although the author acknowledges that the imbalance of DR datasets affects the quality of the generated counterfactuals for underrepresented classes (severe and proliferative diabetic retinopathy), reporting the statistical results for each class will provide a more nuanced understanding of the model's performance and help identify the specialised classes for improvement. Furthermore, as this work aims to generate synthetic retinal images, exploring potential solutions such as data argumentation could deepen the contribution of this study in addressing real-world dataset imbalances.

Also, although the authors provide many visual examples of generated images compared to real images, the introduction of more detailed textual explanations, especially for added or detected visual features in counterfactuals, would enhance the clinical relevance of the work. An audience without a deep background in ophthalmology will benefit from this improvement.

Overall, this manuscript makes a valuable contribution to AI medical image synthesis. The use of diffusion models that combine classifiers to generate clinically meaningful realistic counterfactuals would benefit not only retinal fundus images, but also other medical images. Expert evaluation added credibility to the results. With further analysis and clinical insight, this work has the potential to impact medical imaging and medical education, as well as address real-world challenges such as data set imbalance.

Reviewer #2: Uploaded in word document.

Summary: This is a really nice manuscript that presents a novel AI-based approach to generating realistic and meaningful counterfactual retinal images using diffusion models guided by disease classifiers. The authors propose a method that can illustrate hypothetical scenarios, such as how a healthy fundus image would appear if the subject had diabetic retinopathy (DR). The study validates the approach through a combination of classifier-guided diffusion models and user studies involving ophthalmologists and AI experts. The results and methods here are very useful to the field and have potential for future clinical use, yet require a few more points to be considered before publication.

Major comments:

• Could some of the diseased counterfactuals be generated by changing the blood vessel morphology (e.g., blood vessel density, tortuosity, etc.) in the image rather than just creating lesions? This could potentially explain why you might be seeing barely noticeable lesions in some of the DR DVCs, like discussed in figure 5. Additionally, if these techniques are eventually to be used to help train opthalmologists to diagnose disease, and if DR is diagnosed by means other than just lesions, then it would be good to make sure the images being generated for counterfactual purposes reflect all the ways that DR may be diagnosed. Therefore, I think it would be useful to also test the morphological vessel changes that occur to a disease/healthy generated image. This article (https://www.nature.com/articles/s41467-024-52334-1) seems to show that there are quite a few morphological changes that are coincident with DR.

• The authors indicate that they used publicly available datasets (available online) but the link they shared for eyepacs data only sends you to the eyepacs news announcements part of the eyepacs website. Can you please provide the link to the actual data?

Minor comments:

• Abstract:

o What does it mean to look superficially realistic if experts were able to discern real vs generated CFIs?

• Methods:

o It seems that a single person could have contributed more than one image in the EyePacs dataset, are there concerns that these individuals may be biasing the data as they may be represented more than once (while others may be represented only once)?

o Did the Benitez or FGADR need image quality checks?

o Sentence for “As we require a well-performing..” in section 2.5 seems to have been written incorrectly (grammar is off).

o What defines an AI expert?

• Results:

o What does “GT” stand for in figure 2? It would be useful to include the abbreviations and their corresponding meanings within the respective figure legend.

o There appears to be some speculation in the results that should be left for the discussion.

o Using the phrase “almost impossible to detect” is strong language.

o Is there a way to quantitatively back up the statement: “Upon visual inspection, we found that the diffusion counterfactuals seemed to effectively capture salient features of the various classes.”?

o Should the results in table 5 be corrected for the large differences in the abilities of the ophthalmologist and AI expert in being able to detect the correct image? I would also suggest to consistently call them either AI experts or researchers, and not change between the titles.

Reviewer #3: Summary of key findings: this research introduces an AI algorithm that generates realistic counterfactuals for retinal images using diffusion models, aiming to enhance clinical training and patient education. The algorithm combines diffusion models with classifier guidance to create images showing how a retina might appear under different conditions, such as with or without diabetic retinopathy. The generated images are validated by experts, demonstrating they are clinically meaningful and often indistinguishable from real retinal images. The researchers found that while generated color fundus images were very realistic, the OCT images, while superficially realistic, were easier for experts to identify as AI-generated.

GENERAL COMMENTS

From a technical standpoint, the study appears to be sound, with a well-defined research question and an appropriate methodological approach.

The manuscript under review is well-written and presented in a clear and intelligible manner. The structure of the paper follows a logical progression and figures are clear.

The data presented are consistent with the stated hypotheses, and the conclusions drawn are supported by the findings. Additionally, the code and data used in the research are available on GitHub, with the data sourced from public datasets containing a substantial amount of information. The test sets included over 38,000 healthy images in color fundus photography (CFP) and more than 34,000 healthy images in optical coherence tomography (OCT). This ensures that the findings are well-supported and that further research can build upon these extensive resources.

The code is well-organized and modular, making it suitable for large-scale projects. While the current level of commenting is reasonable, improving inline documentation, especially for complex functions, could enhance maintainability and accessibility for future users.

MAJOR REVISION COMMENTS

• The statement: “we show that we can generate realistic counterfactual generation within the context of retinal disease detection from two retina imaging modalities, CFP and OCT” may need to be revised. Given the authors' own admission that “OCT counterfactuals were relatively easier to detect,” it would be more accurate to restrict the claim to CFP. This clarification would better align the conclusions with the study’s findings.

• The statement: “diffusion probabilistic models have also been investigated for the generation of retinal fundus images [16], however, these methods focus on generation of healthy retinal fundus images” may require reconsideration. There are existing studies on the generation of CFP in disease contexts, such as for Glaucoma (e.g., “Retinal Image Synthesis for Glaucoma Assessment Using DCGAN and VAE Models”). Including a dedicated section in the introduction discussing disease-related generative approaches would strengthen the manuscript. Claims of novelty should be restricted to diseases that have not been previously addressed.

• The statement: “We found that our method generated counterfactuals by introducing or removing the necessary disease-related features as per the query” should explicitly specify that the method targets only diabetic retinopathy (DR). Authors should avoid overgeneralization in similar statements and refer to DR, not disease.

• When restricting their analysis to DR, it seems that their results should be compared with “High-fidelity diabetic retina fundus image synthesis from freestyle lesion maps” (Hou, Biomedical Optics Express 2022). This paper appears to be a direct competitor, and it may deserve mention in the introduction, and direct comparison in Figure 3. Alternatively, the discussion section should explicitly argue the advantages of the proposed method compared to this approach.

6. PLOS authors have the option to publish the peer review history of their article (what does this mean?). If published, this will include your full peer review and any attached files.

**Do you want your identity to be public for this peer review?** For information about this choice, including consent withdrawal, please see our Privacy Policy.

Reviewer #1: No

Reviewer #2: No

Reviewer #3: No

---

## [Decision Letter · Decision Letter 1]

7 Apr 2025

Development and validation of an AI algorithm to generate realistic and meaningful counterfactuals for retinal imaging based on diffusion models

PDIG-D-25-00003R1

Dear Prof. Dr. Berens,

We are pleased to inform you that your manuscript 'Development and validation of an AI algorithm to generate realistic and meaningful counterfactuals for retinal imaging based on diffusion models' has been provisionally accepted for publication in PLOS Digital Health.

Best regards,

Janna Hastings

Guest Editor

PLOS Digital Health

**Additional Editor Comments (if provided):**

**Reviewer Comments (if any, and for reference):**

Reviewer's Responses to Questions

**Comments to the Author**

1. If the authors have adequately addressed your comments raised in a previous round of review and you feel that this manuscript is now acceptable for publication, you may indicate that here to bypass the “Comments to the Author” section, enter your conflict of interest statement in the “Confidential to Editor” section, and submit your "Accept" recommendation.

Reviewer #2: All comments have been addressed

Reviewer #3: All comments have been addressed

2. Does this manuscript meet PLOS Digital Health’s publication criteria? Is the manuscript technically sound, and do the data support the conclusions? The manuscript must describe methodologically and ethically rigorous research with conclusions that are appropriately drawn based on the data presented.

Reviewer #2: Yes

Reviewer #3: Yes

3. Has the statistical analysis been performed appropriately and rigorously?

Reviewer #2: Yes

Reviewer #3: N/A

4. Have the authors made all data underlying the findings in their manuscript fully available (please refer to the Data Availability Statement at the start of the manuscript PDF file)?

Reviewer #2: Yes

Reviewer #3: Yes

5. Is the manuscript presented in an intelligible fashion and written in standard English?

Reviewer #2: Yes

Reviewer #3: Yes

6. Review Comments to the Author

Reviewer #2: No further comments, thank you for addressing my comments.

Reviewer #3: The authors have addressed all of my concerns.

I'm satisfied with their revision and I recommend their paper for publication.

7. PLOS authors have the option to publish the peer review history of their article (what does this mean?). If published, this will include your full peer review and any attached files.

**Do you want your identity to be public for this peer review?** For information about this choice, including consent withdrawal, please see our Privacy Policy.

Reviewer #2: No

Reviewer #3: **Yes: **Mattia Tomasoni
